# Fine-grained Semantic Alignment with Transferred Person-SAM for Text-based Person Retrieval

Yihao Wang
wangyh357@mail2.sysu.edu.cn
School of Computer Science and
Engineering, Sun Yat-Sen University
Guangzhou, Guangdong, China
State Key Laboratory of Integrated
Services Networks, Xidian University
Xi'an, Shaanxi, China
Key Laboratory of Machine
Intelligence and Advanced
Computing (SYSU), Ministry of
Education,
Guangzhou, Guangdong, China

Meng Yang*
yangm6@mail.sysu.edu.cn
School of Computer Science and
Engineering, Sun Yat-Sen University
Guangzhou, Guangdong, China
State Key Laboratory of Integrated
Services Networks, Xidian University
Xi'an, Shaanxi, China
Key Laboratory of Machine
Intelligence and Advanced
Computing (SYSU), Ministry of
Education,
Guangzhou, Guangdong, China

Rui Cao
cr@nwu.edu.cn
School of Information Science and
Technology, State-Province Joint
Engineering and Research Center of
Advanced Networking and Intelligent
Information Services, Northwest
University
Xi'an, Shaanxi, China

## Abstract

Addressing the disparity in description granularity and information gap between images and text has long been a formidable challenge in text-based person retrieval (TBPR) tasks. Recent researchers tried to solve this problem by random local alignment. However, they failed to capture the fine-grained relationships between images and text, so the information and modality gaps remain on the table. We align image regions and text phrases at the same semantic granularity to address the semantic atomicity gap. Our idea is first to extract and then exploit the relationships between fine-grained locals. We introduce a novel Fine-grained **S**emantic **A**lignment with Transferred **P**erson-**SAM** (**SAP-SAM**) approach. By distilling and transferring knowledge, we propose a Person-SAM model to extract fine-grained semantic concepts at the same granularity from images and texts of TBPR and its relationships. With the extracted knowledge, we optimize the fine-grained matching via Explicit Local Concept Alignment and Attentive Cross-modal Decoding to discriminate fine-grained image and text features at the same granularity level and represent the important semantic concepts from both modalities, effectively alleviating the granularity and information gaps. We evaluate our proposed approach on three popular TBPR datasets, demonstrating that SAP-SAM achieves state-of-the-art results and underscores the effectiveness of end-to-end fine-grained local alignment in TBPR tasks.

*Corresponding authors

## CCS Concepts

• **Computing methodologies → Visual content-based indexing and retrieval**.

## Keywords

Text-based Person Retrieval; Visual and Language; Segment Anything

**ACM Reference Format:**
Yihao Wang, Meng Yang, and Rui Cao. 2024. Fine-grained Semantic Alignment with Transferred Person-SAM for Text-based Person Retrieval. In *Proceedings of the 32nd ACM International Conference on Multimedia (MM '24), October 28-November 1, 2024, Melbourne, VIC, Australia.* ACM, New York, NY, USA, 14 pages. https://doi.org/10.1145/3664647.3681553

## 1 Introduction

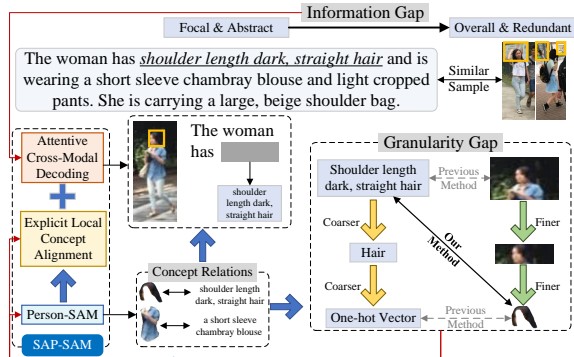

**Figure 1: The granularity and information gaps make it difficult for previous methods to consider the details of fine-grained tasks. Our approach, SAP-SAM, first obtains the relationships of fine-grained features by Person-SAM and then exploits them from two perspectives: fine-grained matching via explicit local concept alignment and informative concept representing via attentive cross-modal decoding.**

**T**ext-**B**ased **P**erson **R**etrieval (**TBPR**) [7, 33, 42, 49] is a fine-grained cross-modal retrieval task, which aims to search for images corresponding to a given textual description in a person gallery. Compared to traditional image-based Person Re-Identification (Re-ID) [11, 19, 53, 60–62, 74, 76], TBPR retrieves images in a cross-modal manner by only requiring text descriptions that are easier to obtain and more lenient, expanding its usage scenarios. As a fine-grained sub-task of image-text retrieval, TBPR focuses on the perception of human beings, one of the most important research objects, benefiting various significant applications of personal albums, surveillance, and public safety.

To realize such a flexible and accurate image-text retrieval algorithm, it is necessary to address the intra-identity variations present in images and textual descriptions, which have been extensively studied in the vision and language communities and have achieved exciting progress. For the fine-grained cross-modal TBPR task, it is much more crucial to address the unique challenges, such as the granularity gap (e.g., as shown in the right of Figure 1, images provide detailed descriptions at the quantitative and pixel level, while textual descriptions are more conceptualized and coarse) and the information gap (e.g., as shown in the top of Figure 1, there are some focuses in the textural descriptions but not in the image).

The granularity gap results in the fineness misalignment of semantic concepts due to the difficulty of extracting the fine-grained feature in the same granularity, preventing the effective cross-modal feature matching between text descriptions and images. Many methods struggle to extract local features (e.g., dividing images into patches straightforwardly [7, 16, 67], using pose estimation [26], performing an auxiliary attribute segmentation [63] and employing sentence analysis tools [43]) or implicitly learn the relationships between cross-modal local parts after extracting their features through deep networks [14, 24, 52, 54, 71, 72]. These methods ignore the granularity gap between text partition (e.g., phrases) and image partition (e.g., image patches). Even if the text is sliced into much smaller units (e.g., words), they still have a larger granularity than an image region corresponding to a specific phrase. For example, a "shirt" is a coarser semantic concept than a specific region of "red wool shirt." The explicit image and text local features in existing methods are not semantically accurate enough, and the implicit alignment methods cannot guarantee that local features, such as image patches and text words, are of the same granularity, preventing further fine-grained alignment learning.

The information gap results in an exact misalignment of semantic concepts due to the characteristic of vision and language, where the former has an overall and redundant visual representation, but the latter has focal and abstract text descriptions. Early studies [32, 33] employed contrastive learning for cross-modal matching, but the information gap is still significant due to the separate encodings of the image and text. The emergence of the vision-language pre-training model has generally alleviated the information gap between images and texts, and variants of contrastive representation learning [17, 37, 73] have been proposed for the task of TBPR. Even through contrastive learning is applied to the cross-modal data, the focal and meaningful semantic concept has not been well extracted from images. The information gap still hinders the effective cross-modal alignment between important semantic concepts of vision and language.

A straightforward idea to overcome these gaps is to extract and align image and text features at the same granularity level, and generate an informative cross-modal attention for exact semantic concepts of vision and language. Following this idea, we proposed a novel Fine-grained **S**emantic **A**lignment with Transferred **P**erson-SAM (**SAP-SAM**) for Text-Based Person Retrieval.

With the idea of distilling knowledge from large pre-trained models (e.g., SAM [28], BLIP [30]) and transferring the knowledge to person retrieval, our approach first captures fine-grained semantic concept relationships in the same granularity from images and texts. To address the granularity gap, we designed Person-SAM, a person segmentation model driven by the text phrase, which is fine-tuned on a human parsing dataset with fine-grained annotations distilled from an off-the-shelf pretrained multi-modal model. Immediately after that, we slice the text in TBPR into fine-grained phrases, and then transfer the trained Person-SAM to extract the segmentation regions corresponding to these fine-grained phrases. It is worth noting that we **only** apply Person-SAM offline in the training phase, and the inference phase is consistent with existing methods.

By exploiting these fine-grained semantic pairs in the same granularity and focusing more on important semantic concepts, we design fine-grained semantic alignment via **E**xplicit **L**ocal **C**oncept **A**lignment (**ELCA**) module and **A**ttentive **C**ross-**M**odal **D**ecoding (**ACMD**) method, respectively. The former pushes the model via cross-modal attention to distinguish whether there is a match between two fine-grained cross-modal concepts. The latter forces the model to predict the selected fine-grained conceptual content with a multi-modal context. We validate our approach on three mainstream benchmarks with state-of-the-art results, and these experiment results demonstrate the effectiveness of our approach. Our contribution consists of the following three points.

1. Using the idea of knowledge distillation and transfer, we propose the Person-SAM model for capturing fine-grained feature relationships in TBPR, effectively alleviating the granularity gap in feature extraction.

2. We design an **E**xplicit **L**ocal **C**oncept **A**lignment (**ELCA**) module to discriminate fine-grained similar features across modalities, effectively alleviating the granularity gap in model learning.

3. We propose an **A**ttentive **C**ross-**M**odal **D**ecoding (**ACMD**) method to understand the important fine-grained context based on information from different modalities, effectively overcoming the information gap.

## 2 Related Work

### 2.1 Global Alignment in TBPR

Early research primarily involved extracting features from texts or images using neural networks, followed by feature aggregation and straightforward cross-modal alignment [7, 32]. Some methods [51, 52, 54] utilize refined single-modality pre-trained models, such as ViT [13], BERT [10], and DeiT [56], at the text and image data, respectively, and align global representations of these models. Others [24, 36] utilize pre-trained multi-modal models, such as CLIP [46], or pre-trained for pedestrian contrast learning, such as UniPT [50], and then fine-tuned on TBPR tasks. However, these

methods need more exact image-text feature interaction or fine-grained cross-modal feature alignment, making it challenging to address the modality gap in TBPR.

Previous research frequently relied on loss functions such as Triple Loss [16], Ranking Loss [27, 41], and Instance Loss [82] for the alignment of text-image representation. However, these methodologies have proven insufficient to address the information-gap and granularity-gap challenges inherent to TBPR. As a response, some researchers [5, 6, 20, 63] have explored innovative approaches that leverage contrastive losses [44]. Additionally, concepts like KL divergence and label smoothing have been introduced by the following studies [23, 24, 79]. While these refinements in loss functions have indeed boosted the performance, they remain predominantly focused on global alignment, which falls short of meeting the demands of fine-grained tasks such as TBPR.

## 2.2    Local Alignment in TBPR

Researchers have used local information for fine-grained alignment. To varying degrees, we classify these methods into two categories: explicit and implicit methods. The former [7, 8, 42, 43, 80] usually possesses obvious correspondences in the data domain, i.e., it pushes an embedding of image part to be similar to an embedding of the text part, which corresponds to meaningful visual regions. Subsequent studies have attempted to utilize hyperpixels like keypoints [12, 29] and additional information like pose [26, 63] for slicing. However, these methods, which tend to be fine-grained at the image level, are not fine-grained at the text level. Meanwhile, due to the limited accuracy and insufficient samples, the image division, hyper-pixels, and pose information are practically challenging to accommodate the multi-scale information in the person concept.

The implicit methods, on the contrary, utilize learnable neural networks to focus on data parts and enhance the local similarity across modalities. Some implicit methods [14, 54, 59, 72] utilize neural networks for local feature correspondence in the embedding space, while still borrowing knowledge (e.g., posture, position, or color) from the explicit data. However, lack of labeling data causes their alignment to be inefficient. Others [24, 51, 52, 64, 71] have adopted a completely implicit feature learning approach, i.e., completely discard this explicit information or prior knowledge. These methods lack a relationship between textual descriptions and image concepts, which causes them to be agnostic. There are also several approaches [15, 24] that borrow ideas from some generalized methods, such as mask language modeling and mask image modeling. However, it is still being determined whether the network in these methods utilizes the corresponding portion of one modality in the reconstruction process of the other modality.

## 3    Method

### 3.1    Overview

We seek a new way to resolve the granularity and information gaps mentioned above between person images and text descriptions. To address the granularity gap, we innovatively learn an enhanced fine-grained feature extraction model with distilling knowledge from SAM and transfer the model to extract fine-grained image-text relationships for TBPR, which are then discriminated in a fine-grained manner across modalities. Moreover, we make the

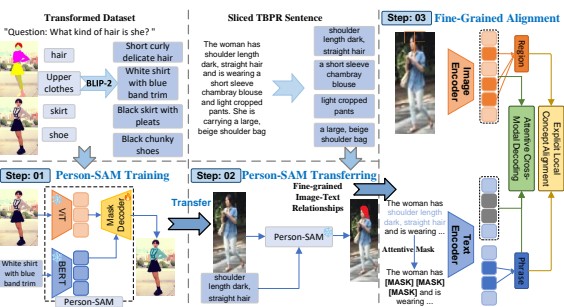

Figure 2: We demonstrate SAP-SAM's workflow. First, we train Person-SAM using the transformed dataset. Then, we obtain the fine-grained image regions corresponding to the sliced TBPR phrases. Finally, we train the retrieval model using these one-to-one relationships.

model focus on the informative image and text semantic concept representation to address the information gap.

Following this idea, we proposed a novel model of Fine-grained **S**emantic **A**lignment with Transferred **P**erson-**SAM** (**SAP-SAM**) for explicitly extracting, aligning and focusing on fine-grained image and text features as shown in Fig. 2. Specifically, we first designed Person-SAM to capture the fine-grained one-to-one relationships between text phrases and image regions, which is the first step. Our Person-SAM is driven by entirely fine-grained texts, not several words or coarse semantic categories, i.e., 'white shirt with blue band trim' rather than 'shirt' or one-hot vector. Due to needing such a fine-grained dataset, we transformed the existing human-parsing dataset for training Person-SAM. In the second step, we transfer the trained Person-SAM to the TBPR dataset according to knowledge distillation and transferring while obtaining image regions corresponding to fine-grained phrases of TBPR. In the third step, we explicitly use these relationships to propose the design of Explicit Local Concept Alignment and Attentive Cross-modal Decoding for these exact local details to push the model's learning of fine-grained semantic concepts, respectively. To the best of our knowledge, we are the first to completely extract and align the fine-grained features of both text and images in TBPR.

Let **I** and **T** denote the image and text modalities, respectively. Given a pair of person image and text **I** and **T**, we further let $S_i$ and $A_i$ denote the $i$-th fine-grained image region and attribute description in the same granularity.

### 3.2    Person-SAM for fine-grained feature extraction

It is very challenging to extract fine-grained semantic concept relationships in TBPR. SAM shows a strong segmentation ability in universal tasks but performs poorly in fine-grained text-region semantics learning of TBPR. Based on SAM, we introduce text encoder and image region prediction modules and realize an end-to-end text-driven segmentation model to directly obtain one-to-one relationships from phrases to regions. This change allows us to extract relationships at the same semantic granularity, simultaneously addressing the granularity gap.

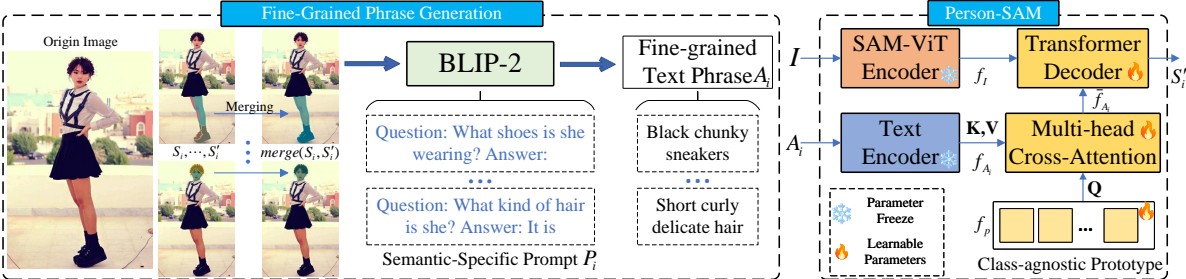

Figure 3: (Left) Fine-grained text description transformations on a human parsing dataset. We use BLIP-2 [30] to generate fine-grained text phrases of the merged semantic regions via textual prompts, constructing fine-grained text-image links. (Right) The structure of Person-SAM. We introduce language models to SAM [28] as textual prompts, and then train on a transformed human parsing dataset. Finally, we will transfer the trained Person-SAM as a knowledge base to generate localized relationships in TBPR.

For training such a fine-grained feature extraction model for TBPR, there are still numerous challenges, one of the most significant being the unavailability of a training dataset. Here, we borrowed the idea of knowledge distillation, transformed the dataset from approximate domains to a usable form, designed Person-SAM, and transferred the trained model to the TBPR training data. As shown in Table 1, we achieved this by adding only a small number of training parameters. Finally, we use only the relationships extracted by Person-SAM in the training phase, while we do not use Person-SAM and any relationships in the validation phase.

**Table 1: Trainable Parameters of our method in different phases.**

| Method | In-context Learning | Fine-Tuning | **Trainable Parameters** |
|---|---|---|---|
| BiLMa[15] | ✘ | ✔ | 220 M |
| BLIP-2 (Section 3.2.2) | ✔ | ✘ | 0 |
| Person-SAM (Section 3.2.1) | ✘ | ✔ | 5 M |
| Alignment (Section 3.3) | ✘ | ✔ | 195 M |

### 3.2.1 Person-SAM Framework.

We first introduce our proposed Person-SAM and how to train it., i.e., **Step: 01** Person-SAM Training. An ideal situation would be if we could have a one-to-one mapping of text phrases to image regions, i.e., $(A_i, \mathbf{I}) \xrightarrow{\theta} S_i$, where $A_i$ is the text prompt, $S_i$ is the segmented image region, $\theta$ is the mapping to be learned. For the implementation of $\theta$, we build a segmentation model based on SAM with text input (that is, $A_i$) instead of coarse class labels (e.g., using "red plaid shirts" instead of "clothing"). **Person-SAM Components.** We utilize the ViT model $\theta_{\text{vit}}$ in vanilla SAM [28] to encode the image $\mathbf{I}$ into the image feature $f_\mathbf{I}$. At the same time, we freeze all parameters of the image encoder. To capture fine-grained text features completely, we discard the dense prompt encoder in vanilla SAM and carefully design the text encoder in our Person-SAM. We keep the decoder $\theta_{\text{dec}}$ in the vanilla SAM, trainable, and decode the segmentation mask corresponding to the text features.

**Text Prompt Design.** Encoding text by introducing Word2Vec [40] can only work for coarse-grained words. Therefore, we introduce a language model, for example, BERT, to enable vanilla SAM to

encode textual phrase prompts, that is, $A_i$. We freeze the parameters of the language model while keeping some learnable parameters $f_p$ after the language model to adapt to the changing fine-grained semantic features in the text. Here, the learnable parameters $f_p$ act as a class-agnostic prototype to push the language model to adapt to the domain-specific information in these phrases. (We describe more details in the Appendix.)

**Image Region Prediction.** As shown in Figure 3, we set a Multi-head Cross-Attention (MCA) layer and make $f_p$ as the query $Q$ and text features as the values $K$, $V$, i.e., $\bar{f}_{A_i} = \theta_{\text{MCA}}\left(f_p, f_{A_i}, f_{A_i}\right)$. Then the image region $S'_i$ predicted from the attribute phrase $A_i$ through the Transformer decoder is denoted as $S'_i = \theta_{\text{dec}}\left(f_\mathbf{I}, \bar{f}_{A_i}\right)$.

Similar to vanilla SAM [28], we also train our Person-SAM by the Dice loss. **However**, we need a fine-grained text-region dataset that can train Person-SAM. Therefore, we considered training Person-SAM by transforming other task datasets of a similar domain.

### 3.2.2 Fine-grained Text Description Generation.
Here, we describe how to transform an existing dataset to train Person-SAM. We propose to generate a fine-grained semantic concept of the person using the available human-parsing dataset[38, 39] and multimodal generative model. However, there are two challenges related to the granularity gap in utilizing the existing dataset and the generative model for generating the fine-grained semantic concept. The first obstacle is that the provided category (denoted by $C_i$ for the $i$-th image region) is coarser than the image region in the human-parsing data. For instance, the category is 'hair' regardless of the hair types of $S_i$, like 'blonde curly' and 'short black hair.' The second obstacle is that the generative model cannot distinguish the granularity of image regions in the human-parsing dataset. For instance, the image region $S_i$ is recognized as 'leg' regardless of the 'left leg' or 'right leg.'

To solve these obstacles, we propose merging the body parts on both sides (e.g., 'left leg' and 'right leg') as a single image region and using the multi-modal generative model to generate the fine-grained attribute description. In addition, we purify this part of the image rather than using key points or other visual prompts for phrase generation. We asked the model about the image content based on the category $C_i$ of the merged image region and designed diverse textual prompts (an example of $P_i$ is: 'Question: What kind

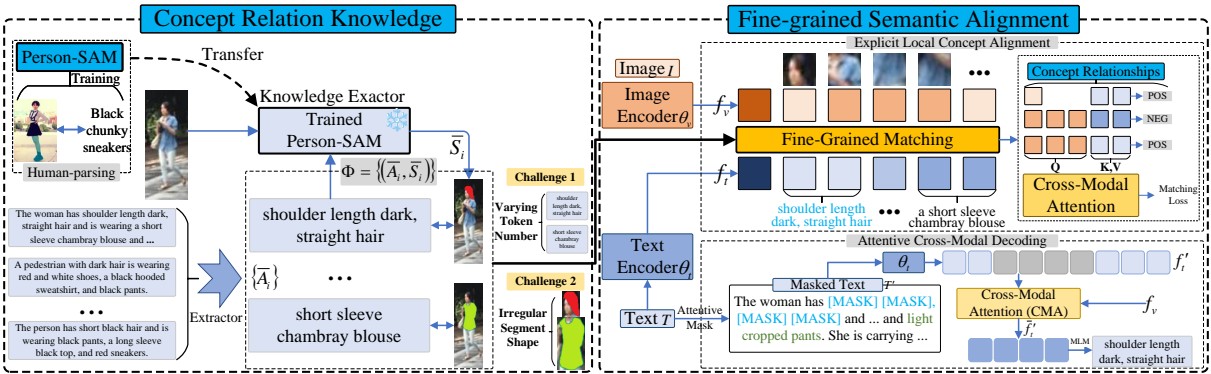

**Figure 4: (Left) We extract the description phrases in TBPR sentences and then input these phrases as text prompts to the transferred Person-SAM to distillate the one-to-one fine-grained concept relationships between textual phrases and image regions. (Right) The Fine-grained Semantic Alignment utilizes the relationships extracted by transferred Person-SAM from two perspectives: discriminant and understanding. The former (Top) enhances the model's discrimination of fine-grained concept relationships, and the latter (Bottom) enhances the model's comprehension of fine-grained content.**

of hat is this? Answer: It is' , **Note:** We describe these details in the Appendix.),

$$A_i = \text{BLIP-2}((\text{merge}(S_i, S_{i'})), P_i) \qquad (1)$$

where merge is the merging operation optional for $S_i$ and its other side $S_{i'}$ and $P_i$ is the predefined textural prompt with $C_i$ for $S_i$, as shown in the left part of Figure 3. $A_i$ is the obtained fine-grained attribute description corresponding to the image region $S_i$.

Finally, we choose BLIP-2 [30] as the generative model. Moreover, we blur the human-parsing image to mitigate the stylistic gap between it and TBPR. In this way, we transformed the human-parsing dataset into an image-text one-to-one fine-grained conceptual dataset for Person-SAM training.

*3.2.3 Concept Relation Extraction.* We then describe how to transfer Person-SAM to TBPR data, i.e., **Step: 02** Person-SAM Transferring. Before extracting fine-grained local correspondences from TBPR data using the pre-trained Person-SAM, it is necessary to extract fine-grained phrase descriptions from the TBPR text. We extract text phrase segmentation from the TBPR text utilizing a simple language model with grammar rules like those of previous methods [43, 52].

For each sentence, as shown in Figure 4, we extract the set of phrases $\{\bar{A}_i\}|_{i=1}^{N_{\text{atr}}}$ corresponding to the TBPR sentence, with $N_{\text{atr}}$ representing the number of phrases. Then, we take these phrases as prompts and feed them into the trained Person-SAM separately, along with the person images $\mathbf{I}$. Finally, we get the local part $\bar{S}_i$ corresponding to each phrase $\bar{A}_i$, that is, $\bar{S}_i = \text{Person-SAM}(\mathbf{I}, \bar{A}_i)$. The outputs of the model are paired as matched fine-grained image-text relationships $\Phi = \{(\bar{A}_i, \bar{S}_i)\}|_{i=1}^{N_{\text{atr}}}$. These concept pairs and the original sample pairs will be used as training inputs. We show some Person-SAM extracted examples in Figure 6.

### 3.3 Fine-grained Semantic Alignment

Finally, we describe how to utilize these relationships, i.e., **Step: 03** Fine-grained Alignment. Obviously, as shown in Fig. 4, aligning the cross-modal fine-grained concept $\Phi$ generated by Person-SAM

still faces two challenges: **Irregular Segment Shape** and **Varying Token Numbers**. The former leads to the problem of representing the image segmentation, while simply aggregating all patches will miss image details. The latter leads to the challenge of representing the text phrase and measuring the cross-modal similarity between local features across modalities.

Unlike existing methods, we propose to solve these challenges by inputting the tokens of image segmentation and text phrase to a cross-modal attention module, in which the cross-modal interaction between image and text will focus on essential parts and generate a fused feature indicating the matching information. This novel approach can deal with irregular image segmentation and different token numbers, effectively aligning the fine-grained local concept across image and text at the same granularity level. Finally, we corrupt those parts of the text that contain full local semantics and push the model to learn the focal and important image representation via reconstructing these parts. Compared to previous approaches, we are fine-grained in the textual perspective, effectively overcoming the information gap.

Denote the image and text encoders by $\theta_v$ and $\theta_t$, respectively. We first use an image encoder to get the image features $f_v = \{v_{[\text{CLS}]}, v_1, \ldots, v_{N_{img}}\}$ of image $\mathbf{I}$, where $v_i$ represents the feature of $i$-th patch in $\mathbf{I}$, where $N_{img}$ represents the patch number. Similarly, we can obtain the text features $f_t = \{t_{[\text{CLS}]}, t_1, \ldots, t_{N_{text}}\}$ of text $\mathbf{T}$, where $N_{text}$ represents the length of the text sequence.

*3.3.1 Explicit Local Concept Alignment.* We propose the **E**xplicit **L**ocal **C**oncept **A**lignment (**ELCA**) module to reduce the granularity gap by pushing the model to distinguish whether fine-grained concepts match. In detail, we first directly obtain the features of each fine-grained semantic concept pair without additional computation. Denote $\mathbf{X}_{\bar{A}_i}$ and $\mathbf{Y}_{\bar{S}_i}$ the indices of the text phrase and image segmentation in the text feature sequence $f_t$ and the image feature sequence $f_v$, respectively. The phrase and image segmentation can then be encoded as $f_t^i = f_t\left[\mathbf{X}_{\bar{A}_i}\right] = \{t_i | i \in \mathbf{X}_{\bar{A}_i}, t_i \in f_t\}$ and

 

$f_v^i = \left\{ v_i | i \in Y_{\bar{S}_i}, v_i \in f_v \right\}$. We then design an explicit local concept alignment task to discriminate fine-grained similar features across modalities, alleviating the granularity gap in model learning.

As shown in Figure 4, we design a cross-modal attention (CMA) module by using image local feature $f_v^i$ as Query, and the text local feature $f_t^j$ as Key and Value

$$f^{i,j} = \text{CMA}(f_v^i, f_t^j, f_t^j) \qquad (2)$$

Let $f_1^{i,j}$ be the first token of the cross-modal attention result. For a semantic concept pair, $(i, j) \in \text{POS}$, $f_1^{i,j}$ is the cross-modal interaction at the same granularity level, indicating the matching information. For the local feature with the index $\bar{i}$ of a given modality, we randomly select another fine-grained localization $\bar{j}$ of the other modality as a negative sample and compute $f_1^{\bar{i},\bar{j}}$, denoted by $(\bar{i}, \bar{j}) \in \text{NEG}$. Then we use a linear classifier, P, to predict whether the fused feature comes from a matched pair

$$\mathcal{L}_{match} = -\sum_{(i,j)}^{\text{POS}} log(\text{P}(f_1^{i,j})) - \sum_{(\bar{i},\bar{j})}^{\text{NEG}} log(1 - \text{P}(f_1^{\bar{i},\bar{j}})) \qquad (3)$$

*3.3.2 Attentive Cross-Modal Decoding.*
To overcome the information gap, instead of using a random mask in cross-modal decoding, we proposed an attentive mask to ensure informative representation in the modalities of image and text.

*Random mask:* In conventional cross-modal decoding [15, 24], a randomized mask is utilized to predict the masked text words from the image. However, random decoding equal for all words would contain many task-irrelevant words (e.g., "is", "this", "he", "and", etc.) that contribute little to the task.

*Attentive mask:* Given the index $\mathbf{X}_{\bar{A}}$ of the text phrases contained in each person description $\mathbf{T}$, the idea is to perturb these task-relevant fine-grained parts more, i.e., the attention mask probability for local concept $p_a$ is much larger than the random mask probability $p_r$ for common words. Denote the attentive mask by $\text{Mask}_a$. By destroying and decoding these key details at $\mathbf{X}_{\bar{A}}$ more frequently, we push the model to learn those semantic details that are truly meaningful,

$$\hat{f}_t = \text{CMA}(\text{Mask}_a(f_t, \mathbf{X}_{\bar{A}}, p_a, p_r), f_v, f_v) \quad \text{s.t.} \quad p_a > p_r \qquad (4)$$

where $\text{Mask}_a(f_t, \mathbf{X}_{\bar{A}}, p_a, p_r)$ is the masked text features and $\hat{f}_t$ is the fused feature using a cross-attention block based on the image features. Finally, the text decoding loss $\mathcal{L}_{dec-t}$ is conducted via the cross-entropy loss between the predicted words from $\hat{f}_t$ and the ground-truth $\mathbf{T}$.

*3.3.3 Optimization Loss Function.* We choose $\mathcal{L}_{SDM}$ proposed by [24] to optimize the global feature similarity of our model, which constrains the similarity of image and text global features to be consistent with the actual distribution through bidirectional KL divergence, i.e., $\mathcal{L}_{SDM} = \text{KL}(\text{sim}_{t2i} || y_{t2i}) + \text{KL}(\text{sim}_{i2t} || y_{i2t})$, where $\text{sim}_{t2i} / y_{t2i}$ and $\text{sim}_{i2t} / y_{i2t}$ represent the text-to-image and image-to-text similarity/truth distributions, respectively. We train our model via minimizing the multi-task loss $\mathcal{L}$ as shown in Equation 5.

$$\mathcal{L} = \mathcal{L}_{match} + \mathcal{L}_{dec-t} + \mathcal{L}_{SDM} \qquad (5)$$

# 4 Experiments
## 4.1 Settings
We validate the performance of the proposed method using publicly available datasets that are frequently used in TBPR tasks, including CUHK-PEDES [33], ICFG-PEDES [12], RSPTReid [84] and ATR Dataset [38, 39]. More datasets information are shown in the Appendix.

## 4.2 Evaluation metrics
Following the established practice, we use the Rank-$k$ metrics ($k = 1, 5, 10$) for evaluation. Given a caption from a query sentence, the model retrieves a corresponding person in the image gallery. If any image of the corresponding person is in the top-$k$ retrieved images, we call it a successful search.

## 4.3 Main Results
We show the results of our method on the three popular benchmarks in Tables 2,3,4, respectively. We train the baseline using Encoder and SDM loss [24].

**Table 2: Main result of our SAP-SAM on CUHK-PEDES.**

| Method | Ref. | Type | R@1 | R@5 | R@10 |
|---|---|---|---|---|---|
| RaSa [1] | IJCAI23 | I. | 57.60 | 78.09 | 84.91 |
| SUM [64] | KBS22 | I. | 59.22 | 80.35 | 87.60 |
| ISANet [72] | arXiv22 | I. | 63.92 | 82.15 | 87.69 |
| SRCF [54] | ECCV22 | I. | 64.04 | 82.99 | 88.81 |
| CM-LRGNet [83] | KBS23 | I. | 64.18 | 82.97 | 89.85 |
| CAIBC [65] | MM22 | I. | 64.43 | 82.87 | 88.37 |
| AXM-Net [14] | AAAI22 | I. | 64.44 | 80.52 | 86.77 |
| LGUR [51] | MM22 | I. | 65.25 | 83.12 | 89.00 |
| IVT [52] | ECCVW22 | I. | 65.59 | 83.11 | 89.21 |
| LCR$^2$S[70] | MM23 | I. | 67.36 | 84.19 | 89.62 |
| VGSG[22] | TIP23 | I. | 67.52 | 84.37 | 90.26 |
| PLIP[85] | arXiv23 | E. | 68.16 | 85.56 | 91.21 |
| UniPT [50] | ICCV23 | E. | 68.50 | 84.67 | 90.38 |
| PDG[68] | TCSVT23 | E. | 69.47 | 87.13 | 92.13 |
| CFine [71] | arXiv22 | I. | 69.57 | 85.93 | 91.15 |
| CSKT[36] | arXiv24 | I. | 69.70 | 86.92 | 91.8 |
| IRRA [24] | CVPR23 | I. | 73.38 | 89.93 | 93.71 |
| BiLMa [15] | ICCV23 | I. | 74.03 | 89.59 | 93.62 |
| **SAP-SAM (Our)** | – | E. | **75.05** | **89.93** | **93.73** |

**CUHK-PEDES** We first evaluate the proposed method on the most common benchmark, CUHK-PEDES. As shown in Table 2, our method outperforms all state-of-the-art methods, achieving 75.05% Rank-1 accuracy, 89.93% Rank-5 accuracy and 93.73% Rank-10 accuracy, respectively. We have characterized how these methods use local features, where "—" means not used with "E" standing for explicit methods and "I" for implicit methods. The results show that our methods surpass previous explicit methods and also recent implicit methods, in particular, 1.67% and 1.02% Rank-1 outperform compared to IRRA [24] and BiLMa [15], respectively. The results indicate that explicitly learning these fine-grained relationships of local features is beneficial. Compared to APTM [75], we are still 1.48% behind on R-1, mainly because APTM built a training dataset of over 1, 500$K$ images and used this large dataset for training, while

CUHK-PEDES only has 34$K$ image data for training (more than 40× times smaller). Nevertheless, even so, our SAP-SAM still leads on R-5 and R-10.

**Table 3: Main result of our SAP-SAM on ICFG-PEDES.**

| Method | Ref. | Type | R@1 | R@5 | R@10 |
|--------|------|------|-----|-----|------|
| SSAN [65] | arXiv21 | E. | 54.23 | 72.63 | 79.53 |
| IVT [52] | ECCVW22 | I. | 56.04 | 73.60 | 80.22 |
| SRCF [54] | ECCV22 | I. | 57.18 | 75.01 | 81.49 |
| PDG[71] | TCSVT23 | E. | 57.69 | 75.79 | 82.67 |
| ISANet [72] | arXiv22 | I. | 57.73 | 75.42 | 81.72 |
| LC$R^2$S[70] | MM23 | I. | 57.93 | 76.08 | 82.40 |
| CSKT[36] | arXiv24 | I. | 58.90 | 77.31 | 83.56 |
| LGUR[51] | MM22 | I. | 59.02 | 75.32 | 81.56 |
| UniPT [50] | ICCV23 | E. | 60.09 | 76.19 | 82.46 |
| VGSG[22] | TIP23 | I. | 60.34 | 76.01 | 82.01 |
| CFine[71] | arXiv22 | I. | 60.83 | 76.55 | 82.42 |
| IRRA [24] | CVPR23 | I. | 63.46 | 80.25 | 85.82 |
| BiLMa [15] | ICCV23 | I. | 63.83 | 80.15 | 85.74 |
| **SAP-SAM (Our)** | — | E. | **63.97** | **80.84** | **86.17** |

**ICFG-PEDES** The performance of the ICFG-PEDES dataset is shown in Table 3. Likewise, the proposed method outperforms all existing methods reported on ICFG-PEDES, achieves Rank-1 of 63.97%, Rank-5 of 80.84% and Rank-10 of 86.17%. We still gained some improvement in all metrics compared to previous randomized [24] or explicit [50] methods.

**Table 4: Main result of our SAP-SAM on RSTPReid.**

| Method | Ref. | Type | R@1 | R@5 | R@10 |
|--------|------|------|-----|-----|------|
| DSSL [84] | MM21 | I. | 39.05 | 62.60 | 73.95 |
| SUM [64] | KBS22 | I. | 41.38 | 67.48 | 76.48 |
| SSAN [12] | arXiv21 | E. | 43.50 | 67.80 | 77.15 |
| LBUL [66] | MM22 | I. | 45.55 | 68.20 | 77.85 |
| IVT [52] | ECCVW22 | I. | 46.70 | 70.00 | 78.80 |
| CAIBC [65] | MM22 | I. | 47.35 | 69.55 | 79.00 |
| CFine [71] | arXiv22 | I. | 50.55 | 72.50 | 81.60 |
| UniPT [50] | ICCV23 | E. | 51.85 | 74.85 | 82.85 |
| LC$R^2$S[70] | MM23 | I. | 54.95 | 76.65 | 84.70 |
| IRRA [24] | CVPR23 | I. | 60.20 | 81.30 | 88.20 |
| BiLMa [15] | ICCV23 | I. | 61.20 | 81.50 | 88.80 |
| **SAP-SAM (Our)** | — | E. | **62.85** | **82.65** | **89.85** |

**RSTPReid** We also validate the effectiveness of our method on a newly proposed dataset, RSTPReid. As shown in Table 4, our method achieves Rank-1 of 62.85%, Rank-5 of 82.65% and Rank-10 of 89.85%, respectively, outperforming previous explicit and implicit methods. In particular, compared to the recently proposed BiLMA [15], a SOTA method, ours outperforms 1.65%, 1.15% and 1.05% the Rank-1, Rank-5, and Rank-10 leads, respectively.

Collectively considering the results of three datasets with different scales, sizes, and focuses, our method outperforms previous methods and achieves state-of-the-art results, once again proving our approach is practical and robust. In contrast to IRRA, we no longer randomize but introduce attentive mask based on the relationships of text phrases, and these results demonstrate the effectiveness of exploiting textual relationships. Compared to BiLMa, we introduce ELCA, and the better results illustrate the importance of

**Table 5: Ablation studies on SAP-SAM components.**

| Method | Components | | | R-1 | R-5 | R-10 |
|--------|:-:|:-:|:-:|-----|-----|------|
| | $\mathcal{L}_{match}$ | $\mathcal{L}_{dec\text{-}t}$ | $\mathcal{L}_{dec\text{-}i}$ | | | |
| Baseline | | | | 70.42 | 86.73 | 92.04 |
| +$\mathcal{L}_{match}$ | ✓ | | | 73.59 | 89.51 | 93.55 |
| +$\mathcal{L}_{dec\text{-}t}$ | | ✓ | | 73.49 | 89.78 | 93.67 |
| +$\mathcal{L}_{dec\text{-}i}$ | | | ✓ | 72.69 | 86.23 | 93.67 |
| +$\mathcal{L}_{match}$+$\mathcal{L}_{dec\text{-}i}$ | ✓ | | ✓ | 74.30 | **90.25** | 93.70 |
| +$\mathcal{L}_{match}$+$\mathcal{L}_{dec\text{-}t}$+$\mathcal{L}_{dec\text{-}i}$ | ✓ | ✓ | ✓ | 74.19 | 89.93 | 93.42 |
| +$\mathcal{L}_{match}$+$\mathcal{L}_{dec\text{-}t}$ | ✓ | ✓ | | **75.05** | 89.93 | **93.73** |

exploiting cross-modal fine-grained conceptual relations, providing new ideas for future methods.

## 4.4 Ablation Study

*4.4.1 Ablations on model components.* As shown in Table 5, the results ('+$\mathcal{L}_{match}$' vs. 'Baseline', '+$\mathcal{L}_{match}$ + $\mathcal{L}_{dec\text{-}t}$' vs. '+$\mathcal{L}_{dec\text{-}t}$', '+$\mathcal{L}_{match}$ + $\mathcal{L}_{dec\text{-}i}$' vs. '+$\mathcal{L}_{dec\text{-}i}$') demonstrate the effectiveness of our proposed ELCA method (Section 3.3.1), with a significant improvement of 3.17%, 2.78%, 1.51%, in all metrics compared to the baseline with the addition of only the fine-grained local alignment method. These results demonstrate the effectiveness of explicitly learning relationships between fine-grained semantics.

For ACMD component (Section 3.3.2), we add both fine-grained text phrases ( '+$\mathcal{L}_{dex\text{-}t}$' vs. 'Baseline', '+$\mathcal{L}_{match}$ +$\mathcal{L}_{dec\text{-}t}$' vs. '+ $\mathcal{L}_{match}$') and fine-grained image regions ('+$\mathcal{L}_{dex\text{-}i}$' vs. 'Baseline', '+$\mathcal{L}_{match}$ + $\mathcal{L}_{dec\text{-}i}$' vs. '+ $\mathcal{L}_{match}$') to the decoding brought significant benefits, yet performance slightly decreased when combining these two approaches ('+$\mathcal{L}_{match}$ + $\mathcal{L}_{dec\text{-}t}$ + $\mathcal{L}_{dec\text{-}i}$' vs. '+$\mathcal{L}_{match}$ + $\mathcal{L}_{dec\text{-}i}$', '+$\mathcal{L}_{match}$ + $\mathcal{L}_{dec\text{-}t}$ + $\mathcal{L}_{dec\text{-}i}$' vs. '+$\mathcal{L}_{match}$ + $\mathcal{L}_{dec\text{-}t}$'). We hypothesize that images contain a more homogeneous information density than text. Text, on the other hand, contains more information only for key phrases due to morphological differences in the data source itself.

*4.4.2 Ablations on Person-SAM relationships.* We used the relationships extracted by Person-SAM on the LGUR[51] and SSAN[12] methods to verify whether these relationships are valid for other methods. The LGUR and the SSAN methods learn fixed fine-grained conceptual divisions, whereas an improvement is obtained when using our relations.

**Table 6: Ablation of Person-SAM Relationships for other Methods on CUHK-PEDES dataset.**

| Method | R@1 | R@5 | R@10 |
|--------|-----|-----|------|
| SSAN [12] | 61.37 | 80.15 | 86.73 |
| SSAN+**Person-SAM** | 62.63 | 81.32 | 87.91 |
| LGUR[51] | 65.25 | 83.12 | 89.00 |
| LGUR+**Person-SAM** | 66.47 | 84.19 | 89.22 |
| Baseline(Our) | 70.42 | 86.73 | 92.04 |
| Baseline+**Person-SAM**(Our) | 75.05 | 89.93 | 93.73 |

As shown in Table 6, the results show that adding the relations extracted by Person-SAM can bring an R-1 improvement of 1.22% to LGUR and an R-1 improvement of 1.37% to SSAN, which demonstrates that our Person-SAM can effectively drive the model's alignment of fine-grained features.

**Table 7: Ablation of Attentive Mask.**

| Method | R@1 | R@5 | R@10 |
|---|---|---|---|
| Random | 72.81 | 89.31 | 93.39 |
| Attentive (Our) | **73.49** | **89.78** | **93.67** |

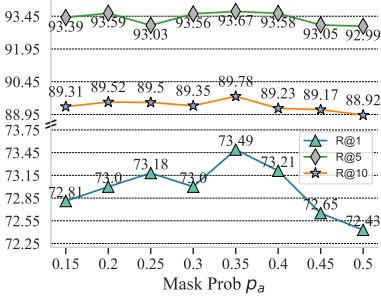

**Figure 5: Ablation study of mask prob $p_a$**

*4.4.3 Ablations on text mask probability.* We explored whether the Attentive mask worked on the CUHK-PEDES dataset. Table 7 compares our original and attentive random masks. This experiment proves that masking the more important parts works in fine-grained tasks. We also explored the effect of the mask probability $p_a$. The results are shown in Figure 5. We started with the probability $p_a = 0.15$ and gradually increased the size of $p_a$ by 0.05 as a step. We obtain the best score when $p_a = 0.35$. Furthermore, performance will drop as $p_a$ increases. We hypothesize that this is due to a high masking probability that can make it difficult for the model to understand the meaning of these key parts of the word, thus impairing performance.

## 4.5 Example of Person-SAM Transferred.

We visualize some segmentation results of Person-SAM, as shown in Fig. 6. The phrases on top of the figure drive these results. Although there are some pixel-level deviations in the Person-SAM produced by TBPR compared to those of the exact image segmentation data, these regions roughly contain the blocks where the objects are present. Person-SAM achieves good results in these examples, whether they are long text phrases (such as the leftmost example at the bottom), small examples (such as the hair, shoes, or bag in the figure), or confusing examples (such as the rightmost example at the top). Admittedly, these results still have some flaws, but they are fine for TBPR.

## 4.6 Case Study

Figure 7 illustrates an example of Top-10 retrieval results for our SAP-SAM and the baseline. As a comparison, our method retrieves more accurate results when the baseline model fails. Further, we found that our approach can capture even small fine-grained semantic concepts. Fine-grained semantic matches can be retrieved even in the wrong cases, as shown by the boxes in the figure. This phenomenon further illustrates the effectiveness of aligning at a fine-grained semantic level and proves that learning the relationship between fine-grained cross-modal semantics is crucial.

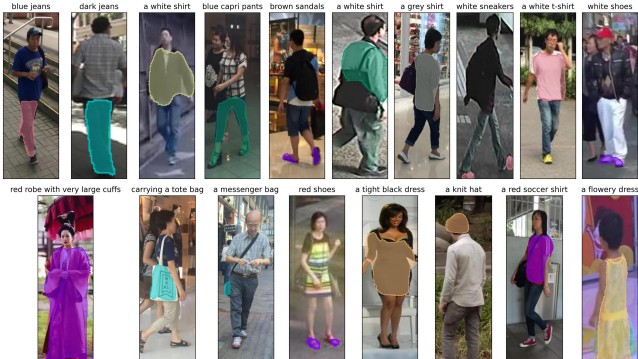

**Figure 6: We randomly show some transferring results of Person-SAM on the TBPR dataset.**

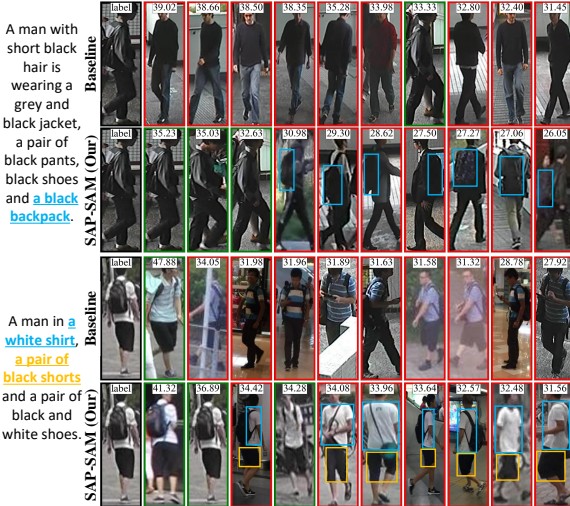

**Figure 7: Retrieval example of our method on the CUHK-PEDES.**

## 5 Conclusion

We propose a novel Fine-grained **S**emantic **A**lignment with Transferred **P**erson-**SAM** (SAP-SAM) method to address the granularity and information gaps in TBPR. First, with knowledge distillation and transferring, we successfully trained the proposed Person-SAM and exactly captured the relationships between fine-grained concepts. Then, in the fine-grained feature matching, our designed **E**xplicit **L**ocal **C**oncept **A**lignment (**ELCA**) effectively enhances the model's discrimination of fine-grained features, addressing the granularity gap in previous approaches. In addition, our proposed **A**ttentive **C**ross-**M**odal **D**ecoding (**ACMD**) has enhanced the model's understanding of fine-grained content, addressing the information gap in previous approaches. We have validated and achieved optimal result on three existing benchmarks, proving our method's effectiveness. The success of utilizing fine-grained cross-modal conceptual relationships also provides new ideas for future research.

# 6 Acknowledgement

This work was supported by the National Natural Science Foundation of China (No. 62176271 and 62072362) and Basic and Applied Basic Research Foundation of Guangdong Province (Grant No. 2024A1515011692).

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

**Table 8: The list of semantic merges performed, with the original semantic categories shown on the left and the right side showing which of the other ones we merged these semantic categories with.**

| Origin | Merged |
|---|---|
| hair | hair, face |
| sun-glass | sun-glass, face |
| left-shoe | left-shoe, right-shoe, left-leg, right-leg |

## A Experiment Details

### A.1 Dataset and Settings

We validate the performance of the proposed method using publicly available datasets that are frequently used in TBPR tasks, including CUHK-PEDES [33], ICFG-PEDES [12], RSPTReid [84] and ATR Dataset [38, 39].

**CUHK-PEDES[33]** is a classical TBPR dataset with $40, 206$ images and $80, 412$ textual descriptions for $13, 003$ identities. The pedestrian image of CUHK-PEDES comes from five existing person re-identification datasets, CUHK03 [35], Market-1501 [81], SSM [69], VIPER [18], and CUHK01 [34]. The training and testing sets comprise $11, 003/1, 000$ persons with $34, 054/3, 074$ images and $68, 108/6, 156$ sentence descriptions, respectively.

**ICFG-PEDES [12]** contains $54, 522$ pedestrian images of $4, 102$ different identities with more fine-grained text descriptions. The images in ICFG-PEDES are collected from the MSMT17 database [67]. The training set contains $34, 674$ image-text pairs from $3, 102$ pedestrians, while the test set contains $19, 848$ image-text pairs for the remaining $1, 000$ pedestrians.

**RSPTReid[84]** is a real scenario text-based person re-identification dataset based on MSMT17 [67]. It contains $20, 505$ images of $4, 101$ persons from 15 cameras in total. The training set consists of $3, 701$ people, $18, 505$ images, and $37, 010$ sentence descriptions. The test set includes $1, 000$ images and $2, 000$ textual descriptions of 200 pedestrians.

**ATR Dataset[38, 39]** is a human parsing dataset with $17, 706$ images and 18 semantic categories. The images in ATR come from diverse sources. Each image is assigned several semantic categories and labeled with fine pixel-level annotations. We randomly split the ATR dataset into the train, valid, and test sets with the ratio of $8 : 1 : 1$.

### A.2 Implementation Details

We loaded the pre-trained CLIP-B/16 for the text and image encoders and randomly initiated the rest of the modules. During training and testing, all images are uniformly scaled to $384 \times 128$, and the maximum length of the text is set to 77. We train the model using the Adam optimizer and set the learning rate of the pre-training module to $1e$-5 with the cosine decay strategy. For the other modules, we set the learning rate $5e$-5. The mask probability $p'$ is set to 0.35. The size of the image patch is 16. For the training of the segmentation model, we use SAM-Base and BERT-base-uncased for the text model. We freeze the base encoder and train only the decoder and other parameters. We set the learning rate to $1e$-4, and

train 20 epochs. The image input size of the model is $1024 \times 1024$, and the maximum length of each phrase is 16.

## B Details of Person-SAM Tuning

### B.1 Semantic Merging in Person-SAM

We performed a semantic merging on the ATR dataset when training Person-SAM in the main text, and here we explain in detail why we conducted this operation. First, the original ATR data consists of 18 categories (including the background). At the same time, it contains, for example, categories with positional information such as "left-shoe" and "right-shoe," as shown in Fig. 8. These categories usually appear in entirely different shapes due to the masking of the parts. As shown in the three left columns of Figure 8, we demonstrate some masked semantic classes that are unfavorable for training text-driven semantic segmentation. Therefore, we need to merge these similar semantics to mitigate some of the effects due to masking. In addition to this, for example, as shown in the two right columns of Figure 8, smaller regions, such as glasses, are challenging to achieve detailed generation of linguistic descriptions when used alone, whereas the generative model can work well when merged with the face. Finally, as shown in Table 8, we will merge the semantic classes. Instead of merging, we generate the phrases corresponding to these regions separately for the rest of the semantic classes.

In addition to this, inspired by some recent research on visual prompts[9, 57, 77], we still tried to utilize some visual prompts to directly generate descriptions of the corresponding regions, as shown in Fig. 9. We attempted to use visual prompts for points, regions, and boxes, respectively, and asked for the content of the corresponding regions via text. However, this result was not satisfactory, and these attempts invariably produced very noisy results while still generating many errors when confronted with those fine semantic classes. Therefore, we directly filter as much irrelevant interference as possible during the actual generation process and target the prompts for each semantic class.

### B.2 Text Prompt for Phrase Generation

After merging the semantic classes and processing the image regions, we next need to design the textual prompts to make the model output fine-grained phrases as correctly as possible. Some existing research also suggests that textual prompts[3, 4, 25, 45, 47, 55], even some punctuation marks that seem small to humans, may contain very unique semantic information to the model. Therefore, we need to design these linguistic prompts very carefully. We take a natural language perspective, keeping sentences as fluent as possible, and ask questions about the current semantic class. This operation aims to narrow down the model's choices of what to output so that the model outputs more accurate results than without containing the current semantic class word. Taking hair as an example, we show the design of some semantic classes, as shown in Table 9. Finally, we manually evaluate the generation quality of each text prompt on several samples and select the most accurate one for the template. We perform this operation for each semantic class, and finally, we get the specific text prompt templates for all semantic classes as shown in Table 10.

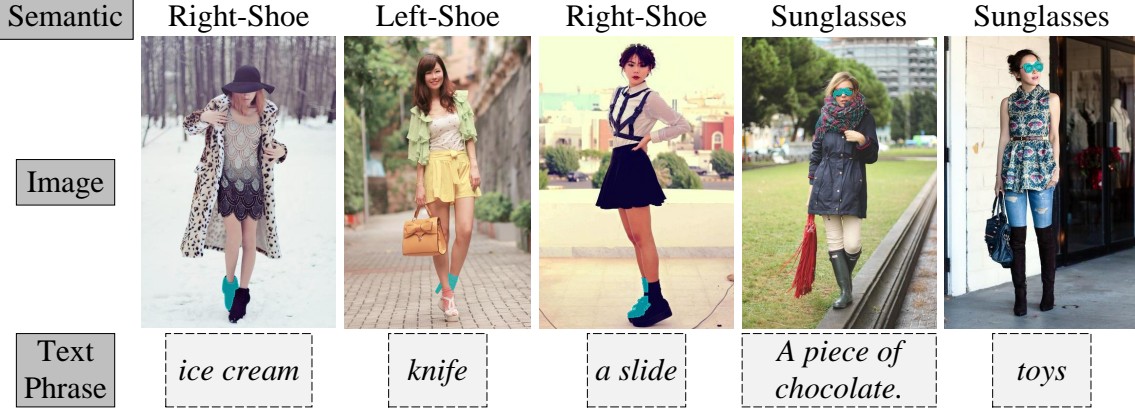

**Figure 8: When the object is occluded or very small, the generated model can easily misclassify it. The three columns on the left show examples where BLIP-2 produced an error after the object occluded. The two columns on the right show examples where the generated results are erroneous when the objects are very small.**

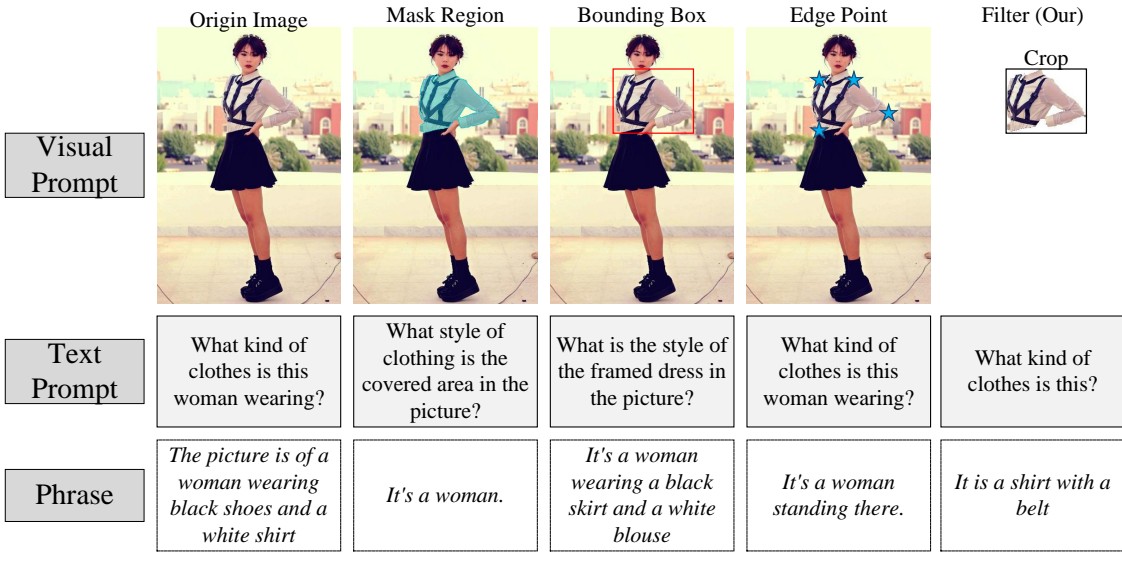

**Figure 9: From left to right, visual region processing includes the original image (prompted by text only), mask enhancement, localization box enhancement, keypoint enhancement, and irrelevant region filtering. In several other methods, more or less irrelevant or erroneous bits of content are produced.**

## C Ablations on Image Mask

Section 4.3 of the main text ablates the attentive cross-modal decoding module. We include an $\mathcal{L}_{dec\text{-}i}$ that needs to be mentioned in the main text. The $\mathcal{L}_{dec\text{-}i}$ stands for masking the image features and then, in a similar way to the text, improving the model's understanding of the fine-grained features by introducing cross-modal reconstruction. Unlike text, however, current research is still not unified on how to model masked images, so we have tried two main types of approaches: discrete (as used by BeiT [2], for example) and linear (as used by MAE [21], for example).

(1) For discrete, we follow BeiT and quantize the image using a pre-trained VQ-VAE[58] (we chose the DALLE[48] pre-trained

one), i.e., ids = VQ-VAE($\mathbf{I}$). Then, similarly to text, we mask the image patches to obtain the masked features $f_v'$. Next, we decode these masked tokens using the textual features $f_t$, i.e., $\bar{f}_v' = \text{MCA}(f_v', f_t, f_t)$. Finally, similarly to text, we find these masked parts and then predict the ids that these features initially corresponded, that is, $\mathcal{L}_{dec\text{-}i} = \ell_{CE}(\text{MLP}(\bar{f}_v'), \text{ids})$.

(2) During the experiment, we find that a more significant portion of the vectors acquired by VQ-VAE belong to the background (about 31%, i.e., $\text{ids}_i = 0$). To avoid the category imbalance problem, we ignore these backgrounds and reconstruct only the other tokens $\text{ids}' = \{x \neq 0 | x \in \text{ids}\}$, i.e., $\mathcal{L}_{dec\text{-}i} = \ell_{CE}(\text{MLP}(\bar{f}_v'), \text{ids}')$.

**Table 9: Ablation studies on Text Prompt for BLIP-2.**

| Prompt | Response |
|---|---|
| **None** | A black shell. |
| What object is in this picture? Answer: | This picture has a piece of dark chocolate. |
| What's in this picture? Answer: | There's a headset in the picture |
| Question: What's her hairstyle? Answer: | short |
| Question: What kind of hair she has? Answer: | She has short, curly hair. |
| Question: What kind of hair is this? Answer: | The kind of hair she has is short. |
| Question: What kind of hair is she? Answer: She has | short, curly hair |

**Table 12: Results of Different Person-SAM Text Prompt Methos.**

| No. | Prompt Feature | mIoU |
|---|---|---|
| 1) | $f_{A_i}^{[\text{CLS}]}$ | 56.31 |
| 2) | $f_{A_i}$ | 59.27 |
| 3) | $f_{A_i}^{[\text{CLS}]} \cdot f_{\text{I}}$ | 58.01 |
| 4) | $\theta_{\text{MCA}}\left(f_p, f_{A_i}, f_{A_i}\right)$ | 61.55 |

**Table 10: Prompts corresponding to all semantic classes used for BLIP-2.**

| Semantic | Prompt |
|---|---|
| hat | Question: What kind of hat is this? Answer: It is |
| hair | Question: What kind of hair is she? Answer: She has |
| glasses | Question: What kind of glasses is this? Answer: It is |
| clothes | Question: What kind of clothes is this? Answer: It is |
| skirt | Question: What kind of skirt is this? Answer: It is |
| pants | Question: What kind of pants is this? Answer: It is |
| dress | Question: What kind of dress is this? Answer: It is |
| shoes | Question: What shoes is she wearing? Answer: She is wearing |
| bag | Question: What kind of bag is this? Answer: It is |

**Table 11: Ablations on Image Decoding.**

| No. | Method | R-1 | R-5 | R-10 |
|---|---|---|---|---|
| 1) | discrete | 69.54 | 85.21 | 92.65 |
| 2) | discrete w/o background | 69.77 | 85.63 | 92.74 |
| 3) | **linear & after PE** | **72.69** | **86.23** | **93.67** |
| 4) | linear & after ViT | 71.56 | 85.97 | 93.01 |

We also have two strategies for linear features: (3) First, the image **I** is masked immediately after Patch Embedding and the masked tokens are fed to the subsequent attention layer in ViT.

(4) The other is to mask the image features $f_v$ acquired by ViT. Unlike the discrete strategy, both strategies use MSE loss $\ell_{MSE}$ to compute the decoding loss $\mathcal{L}_{dec\text{-}i}$. In addition to that, the rest of the masking and decoding strategies remain unchanged.

The results of these strategies are shown in Table 11, with strategy **(3)** achieving the best results. Although image cross-modal decoding alone is effective, performance impairment occurs when combining text cross-modal decoding with image cross-modal decoding, as shown in the main text. How to solve this problem is also part of our subsequent research.

## D  Discussion of Person-SAM Structure

In the design of Person-SAM, we introduced a text encoder to accommodate text-driven prompts to generate exactly corresponding fine-grained regions. We explored a few ways to use the text features, and the segmentation result in the ATR dataset is shown in Table 12, and we used mAP as the evaluation criterion.

We then briefly describe these methods. **Method** $f_{A_i}^{[\text{CLS}]}$ means using the [CLS] token feature of the text $A_i$ as text prompt. **Method** $f_{A_i}$ meas using dense text features (i.e., features of all $A_i$ tokens. **Method** $f_{A_i}^{[\text{CLS}]} \cdot f_{\text{I}}$ means using dot products between dense textual prompts with image features as prompts for the decoder. **Method** $\theta_{\text{MCA}}\left(f_p, f_{A_i}, f_{A_i}\right)$, which we used, means using Multi-head Cross-Attention(MCA) to obtain the fused textual features between some learnable tokens $f_p$ and the dense textual features as input to the decoder's prompts.

As shown in Table 12, method $\theta_{\text{MCA}}\left(f_p, f_{A_i}, f_{A_i}\right)$ is better than the other methods, so we use this method in the Person-SAM structure. The reason for this phenomenon is that fine-grained features require an exact representation, so methods with dense features $f_{A_i}$ are better than methods using $f_{A_i}^{[\text{CLS}]}$. At the same time, the learnable parameters provide domain adaptation, which helps Person-SAM to focus more easily on details related to pedestrians.

## E  Discussion on Alignment Strategy.

In addition to the alignment strategies described in Section 3.3.1, inspired by previous work, like GLIP[31], X-VLM[78], we still explored some other alignment strategies. **Method** 'avgPool' means using average pooling to aggregate each local feature, i.e., $\bar{f}_v^i = \text{avgPool}\left(\hat{f}_v^i\right)$ and $\bar{f}_t^i = \text{avgPool}\left(\hat{f}_t^i\right)$, and then employing the InfoNCE loss to constrain the cosine similarity between image and text local features. **Method** 'Conv1D' means using 1-$d$ convolution network instead of average pooling in method 'avgPool' with

**Table 13: Results of Different Alignment Strategy.**

| No. | Method | R-1 | R-5 | R-10 |
|-----|--------|-----|-----|------|
|  | w/o matching | 70.42 | 86.73 | 92.04 |
| 1) | avgPool | 32.96(↓) | 55.10 | 65.53 |
| 2) | Conv1D | 53.87(↓) | 76.49 | 83.35 |
| 3) | MHSA | 68.99(↓) | 87.33 | 92.07 |
| 4) | **ELCA (Our)** | **73.59** | **89.51** | **93.55** |

the rest remaining unchanged. **Method** 'MHSA' means utilizing a multi-head self-attention (MHSA) module to aggregate text and image features, i.e., $\bar{f}_v^i = \text{MHSA}\left(\left[\bar{v}, \hat{f}_v^i\right]\right)$ and $\bar{f}_t^i = \text{MHSA}\left(\left[\bar{t}, \hat{f}_t^i\right]\right)$, then using the InfoNCE loss to constrain the cosine similarity between the [CLS] feature of $\bar{f}_v^i$ and $\bar{f}_t^i$, where $\bar{v}$ and $\bar{t}$ are learnable parameter token, $[\cdot]$ is concatenated operation. **Method** 'ELCA' is our explicit local concept alignment method described in Section 3.3.1.

Table 13 shows the influence of different matching approaches. The benefit of retaining features for all tokens compared to aggregated features, such as 'avgPool', suggests that the aggregated features lose some fine-grained information that is trivial in traditional tasks but critical in TBPR. Our ELCA can push the model more strongly to discriminate semantic differences between localizations by interacting with fine-grained features. This result demonstrates the specificity of the TBPR task, i.e., it is a fine-grained task, and the importance of aligning fine-grained semantic features.

## F Scaling on larger model

**Table 14: Main result of our SAP-SAM using larger backbone on CUHK-PEDES.**

| Method | Backbone | R@1 | R@5 | R@10 |
|--------|----------|-----|-----|------|
| Baseline | CLIP (ViT/B-16) | 70.42 | 86.73 | 92.04 |
| **SAP-SAM (Our)** | CLIP (ViT/B-16). | **75.05** | **89.93** | **93.73** |
| Baseline | CLIP (ViT/L-14) | 72.13 | 87.15 | 92.71 |
| **SAP-SAM (Our)** | CLIP (ViT/L-14). | **76.28** | **90.87** | **94.75** |

We still trained SAP-SAM on a larger backbone network, such as CLIP(ViT/L-14), and the results are shown in Table 14. Our SAP-SAM achieved better results, but we did not use this result in the text for a fair comparison.

## G Limitations

We mainly propose a fine-grained local feature alignment method for images and text to improve the quality of the model's representation of cross-modal features through fine-grained feature identification and understanding, thus improving the model's performance in downstream tasks. We focus on some problems in the TBPR task from a fine-grained perspective. However, due to resource constraints, our approach still has some problems:

- In Person-SAM transfer, due to the limited computational resources, the model we chose is small, which may limit some of the model's capabilities and lead to less fine-grained results obtained.
- Since there are still some domain differences between the ATR and TBPR datasets in the transfer process, this problem still exists even though we have performed some data style transformations.
- Since we retained all the feature blocks, the computation process took up more time during the learning process of fine-grained features.

In the future, we will also investigate how to learn this relationship faster.