# OpenReview forum: "Fine-grained Semantic Alignment with Transferred Person-SAM for Text-based Person Retrieval"
_acmmm.org/ACMMM/2024/Conference — MM2024 Poster_

### Official Review · Reviewer_Hf5n · 2024-05-20

**Rating:** 4
**Confidence:** 4

**Summary:**

The paper introduces a Fine-grained Semantic Alignment with Transferred Person-SAM (SAP-SAM) model to address the granularity gap in text-based person retrieval (TBPR). The Person-SAM model is fine-tuned on a human parsing dataset using fine-grained annotations derived from an off-the-shelf pre-trained multi-modal model. Text in TBPR is segmented into fine-grained phrases, and these segments are aligned with corresponding segmented areas in images using the trained Person-SAM model. The alignment focuses on semantic pairs of the same granularity, emphasizing important semantic concepts through Explicit Local Concept Alignment (ELCA) and Attentive Cross-Modal Decoding (ACMD). ELCA uses cross-modal attention to distinguish matches between fine-grained cross-modal concepts, while ACMD predicts fine-grained conceptual content within a multi-modal context. The approach is validated on three mainstream benchmarks, demonstrating its effectiveness with state-of-the-art results. This method significantly improves the granularity and information alignment between images and text in TBPR tasks.

**Strengths:**

The paper is grammatically fluent and structurally clear, and the ablation study provided is very detailed. Furthermore, unlike previous methods that employed random alignment, this paper innovatively uses pre-trained multi-modal models to add fine-grained annotations to a human parsing dataset for fine-tuning the Person-SAM model. This approach enhances the model's ability to accurately segment fine-grained textual phrases and align them with corresponding image regions, thus significantly improving the alignment capabilities between text and images.

**Limitations:**

1. Figure 1 has a spelling error: 'Overall & Redudant' should be corrected to 'Overall & Redundant'.

2. The APTM method mentioned in line 673 of this paper also includes a cross-modal attention matching module and masked language modeling. Given the inconsistency in training data scale, a direct and fair performance comparison cannot be conducted. Therefore, could the Person-SAM model be added on top of the APTM framework to facilitate a performance comparison with APTM? Moreover, it seems that the performance is not as stated in the text: 'Compared to APTM [58], we are still 1.12% behind on R-1, … Nevertheless, even so, our SAP-SAM still leads on R-5 and R-10.' In fact, this paper reports being 1.48% lower on R-1 than APTM, and also lower on R-5 and R-10.

3. Although the Person-SAM is only applied offline during the training phase, using this model along with fine-grained semantic alignment could potentially result in higher computational costs. The supplementary materials mention that fine-grained semantics consume more computational resources; however, they do not provide specific details on the model's runtime and the computational resources required, which are critical considerations for practical applications.

4. During the inference phase, it is necessary to input text features and image features in pairs into the cross-modal attention to obtain matching scores. This seems to require sorting the features output by the encoder, followed by selecting the top-k pairs of images and texts for input. The paper does not mention the specific implementation process for this procedure.

**Suitability:**

3

---

### Official Review · Reviewer_LwPF · 2024-05-21

**Rating:** 4
**Confidence:** 4

**Summary:**

This paper proposed a novel approach called Fine-grained Semantic Alignment with Transferred Person-SAM (SAP-SAM). SAP-SAM addressed the disparity in description granularity and information gaps between images and text in TBPR tasks by extracting and aligning fine-grained textual and visual features using a transferred Person-SAM model, and exploiting the captured relationships via Explicit Local Concept Alignment (ELCA) and Attention Cross-Modal Decoding (ACMD) modules to enhance fine-grained discrimination and cross-modal understanding. SAP-SAM was evaluated on CUHK-PEDES, ICFG-PEDES, and RSTPReid, outperforming SoTA methods. Ablation experiments demonstrate the effectiveness of the proposed components.

**Strengths:**

1. By fine-tuning the person segmentation (Person-SAM) model on human parsing datasets, it learns to capture fine-grained relationships between text phrases and image regions at the same semantic granularity.
2. The ELCA module enhances the model's ability to distinguish matched fine-grained concepts across modalities, which improves fine-grained discrimination.
3. The ACMD focuses on informative semantic concepts by using an attentive mask. This helps the model understand the important fine-grained context based on information from different modalities.
4. SAP-SAM leverages the idea of knowledge distillation and transfer to train the Person-SAM model on existing datasets. This allows the model to learn from related domains and improves its ability to capture fine-grained relationships in TBPR data.

**Limitations:**

1. The authors acknowledge pixel-level deviations compared to the exact image segmentation data. These deviations may affect the precision of the captured fine-grained relationships.
2. The authors compare SAP-SAM with the APTM method, which uses a large-scale pre-training dataset of over 1,500K images. In contrast, the CUHK-PEDES dataset for training SAP-SAM has only 34K images. While SAP-SAM still outperforms APTM on Rank-5 and Rank-10 metrics, it falls short on Rank-1 accuracy. This suggests that large-scale pre-training could potentially further improve SAP-SAM’s performance.
3. While the authors emphasize that Person-SAM is only used offline during training, the computational overhead of this process is not explicitly discussed.
4. The interpretability of the learned relationships and SAP-SAM’s decision-making process is not extensively discussed in the paper.

**Suitability:**

3

---

### Official Review · Reviewer_x7qJ · 2024-05-24

**Rating:** 5
**Confidence:** 4

**Summary:**

This paper proposes a Fine-grained Semantic Alignment model with Transferred Person-SAM (SAP-SAM) to effectively address the granularity and information gaps inherent in Text-based Person Retrieval task.

**Strengths:**

1. This paper reveals the necessity of explicit alignment and fine-grained alignment, constructs fine-grained annotations for text-based person retrieval task and proposes a Person-SAM model to capture fine-grained feature relationships in TBPR.
2. This paper proposes the ELCA and ACMD modules to exploit the relationship of fine-grained semantic concepts in the same granularity.

**Limitations:**

1. When the fine-grained text phrases are obtained, the proposed Person-SAM can be replaced by the human-parsing model proposed by VITAA[1]. If so, only the inference step is needed to construct fine-grained region-phrase pairs, and there's no need to train the model.
2. Why not do the fine-grained semantic alignment directly by pretrained Person-SAM instead of designing the ELCA module?
3. Implementation details should be included in the main text rather than in the appendix.
4. There are many logical inconsistencies and unclear causal expressions in the paper. (For instance, L156, L307, L375, L505)

[1] Zhe Wang, Zhiyuan Fang, Jun Wang, and Yezhou Yang. 2020. Vitaa: Visualtextual attributes alignment in person search by natural language. In Computer Vision–ECCV 2020: 16th European Conference, Glasgow, UK, August 23–28, 2020, Proceedings, Part XII 16. Springer, 402–420.

**Suitability:**

3

---

### Official Review · Reviewer_5ivn · 2024-05-26

**Rating:** 4
**Confidence:** 3

**Summary:**

1、This paper primarily addresses the issue of the granularity gap between textual descriptions and real image information in text-based person retrieval tasks by aligning text phrases and images at the same semantic granularity to bridge the semantic gap.
2、To solve this problem, the authors first propose a method called Person-SAM (SAP-SAM) to align fine-grained image and text phrases. They extract knowledge from SAM through a pre-training approach and transfer it to extract fine-grained image-text relationships using the TBPR task format dataset. During training, they optimize fine-grained matching through explicit local concept alignment and attention-based cross-modal decoding, distinguishing fine-grained image and text features at the same granularity level.

**Strengths:**

The paper innovatively proposes a new text-image retrieval method based on fine-grained semantic alignment. By fine-tuning the transferred knowledge from the SAM large model to adapt it to the TBPR task, the Person-SAM structure is proposed. To address two shortcomings of Person-SAM, the Explicit Local Concept Alignment (ELCA) and Attention-based Cross-Modal Decoding (ACMD) modules are introduced. The effectiveness of the experiments is validated on three benchmark datasets.

**Limitations:**

1. The method proposed in this paper consists of a pre-training stage and a training stage. What about the training time complexity?
2. The experimental results of the RaSa method in Table 2 differ significantly from the original paper, while the reason is not explained. Also no comparison of the experimental results of the RaSa method in Tables 3 and 4. Please supplement the experiments or provide an explanation.
3. The reason for the lower comparison results between this method and the APTM method as stated is that APTM proposed a new large-scale dataset for pre-training. This argument is not fully substantiated. Can this method also be pre-trained on the large-scale dataset proposed by APTM?
4. The impact of the Explicit Local Concept Alignment (ELCA) and Attentive Cross-Modal Decoding (ACMD) modules on the final validation results lacks ablation experiment verification.
5. The experimental data in the method diagram of the article is too extensive and can be appropriately omitted.

**Suitability:**

2

---

### Meta-Review · Area_Chair_ujnX · 2024-07-11

**Recommendation:** Accept (Poster)
**Confidence:** 3

**Metareview:**

All the reviewers recognize that the technical contribution and experimental results are acceptable.
This submission is acceptable if room